# Metrology Benchmarking of 3D Scanning Sensors Using a Ceramic GD&T-Based Artefact

**DOI:** 10.3390/s22228596

**Published:** 2022-11-08

**Authors:** Eduardo Cuesta, Víctor Meana, Braulio J. Álvarez, Sara Giganto, Susana Martínez-Pellitero

**Affiliations:** 1Department of Construction and Manufacturing Engineering, University of Oviedo, Campus de Gijón, 33203 Gijón, Spain; 2Area of Manufacturing Engineering, University of León, Campus de Vegazana, 24071 León, Spain

**Keywords:** 3D optical scanner, benchmarking, metrological evaluation, GD&T-based artefact, reverse engineering

## Abstract

The use of non-contact scanning equipment in metrology and in dimensional and geometric inspection applications is increasing due to its ease of use, the speed and density of scans, and the current costs. In fact, these technologies are becoming increasingly dominant in the industrial environment, thus moving from reverse engineering applications to metrological applications. However, this planned transfer requires actions to ensure the achievable accuracy by providing traceability of measurements. In the present study, a comparison between the devices is carried out and a specific standard artefact is designed, equipped with multiple ceramic optically friendly entities, and allowing a wide variety of geometric dimensioning and tolerancing (GD&T). Four different 3D scanning sensors are used in the experimentation. Three of them are based on laser triangulation, and the fourth is a structured blue light sensor (fringe pattern projection). The standard artefact is calibrated with a high accuracy, using a coordinate measuring machine (CMM) and probing sensors. With this CMM, reference values of multiple predefined GD&T are obtained. The evaluation methodology maximises the accuracy of each device in measuring the dimensions of the artefact due to the good dimensional (milling and turning), surface (control of machining variables), and the dimensional and spatial distribution characteristics. The procedure also includes the same treatment of the captured point clouds (trimming, filtering, and best-fit algorithm, etc.) in each of the four 3D scanning sensors considered. From this process, very reliable measurements of the maximum achievable accuracy of each device (deviations from the CMM measurements) are finally obtained, and a multi-characteristic comparison between the four sensors is performed, also with high reliability.

## 1. Introduction

Currently, industries demand the procedures and artefacts that guarantee, in a simple way, the traceability of measurements performed in the field of dimensional inspection [1]. This is even more important in the Industry 4.0 context, where the aim is to digitise not only the manufactured products, but also the machines themselves, as well as the tools and fixtures, etc., and the processes in general.

For this reason, non-contact metrology inspection and reverse engineering equipment are essential. As evidence of this, these two aspects are undergoing an unprecedented industrial deployment that is not only due to their increasingly wide field of application, but also to the efforts of the manufacturers themselves in terms of performance enhancement and price containment. However, it is obvious that at present these two characteristics (performance and cost) are not incompatible, being not as far apart from each other as they were ten years ago. Today, 3D scanners and reverse engineering equipment are already available on the market with accuracies of less than 0.1 mm and at quite a reduced price (even below 3000 USD). Moreover, they have a high capacity, in terms of point cloud density, even allowing point capturing on black surfaces, which was impossible a few years ago, at least with an acceptable precision.

Unfortunately, as is often the case with the latest generation of developments in technology, the accuracy offered by manufacturers is difficult to verify, and even more difficult to compare between different technologies. It should be noted that there are very different groups of non-contact dimensional inspection technologies, and even their terminology is under constant review [2]. On the one hand, there are systems based on direct range detection (Laser Tracker Time of Flight, Phase shift, Continuous-Wave Radar, Flash LIDAR, and 3D camera range or Laser Radar, etc.). On the other hand, there is equipment wherein the light source and the capture system are not collinear, but present a certain angle between them (triangulation-based devices). This occurs in the case of laser triangulation sensors or structured light devices (also known as white/blue light scanners and fringe projection scanners, etc.), and photogrammetry, etc. In addition, there are other different technologies such as those that work with colinear light beams (or laser) at smaller ranges (micro and nanoscale), such as conoscopic holography, focus variation, and optoelectronic profilometers, etc.

It is evident that the procedures and artefacts developed to verify the performance of non-contact equipment are clearly different depending on whether they are used in metrological or reverse engineering applications. In the first case, artefacts of small dimensions are used, usually a precision sphere or a set of them [3]. However, devices for reverse engineering are applied to different purposes: preservation of heritage, art or archaeology [4], repairing of industrial components [5], replication of models or prototypes for mould production (automotive sector) [6], and positioning and alignment of medium and large parts in the assembly of machines and structures (aeronautics, buildings), etc. Such equipment requires artefacts of very different sizes as well as accuracy from those artefacts used for verification and/or calibration in metrological applications. Two dimensional artefacts, based on targets or circles on boards, checkerboards with squares, or ad-hoc artefacts are often used In contrast, in metrological applications of 3D scanners, as it is often the case with triangulation lasers mounted on CMMs or AACMMs [7,8,9], accuracy takes precedence over scanning density and speed, especially when there are implications for dimensional and geometric tolerance (GD&T) compliance. An example of this is the digitising applied to the verification of additive manufactured parts, which, when finally machined, have multiple mechanical applications in the automotive sector (components and tooling), or the medical sector (implantology and prostheses). In this context, comparisons between equipment [10,11], verification and/or recertification procedures [12], and the development of artefacts [1] become much more relevant. In this study a new artefact is proposed, which is specifically designed to verify the performance of different types of non-contact 3D scanners in mechanical applications with GD&T specifications.

There are many types of 3D scanners with different technologies and fields of application, although with certain overlaps between them and, therefore, with very disparate calibrations and/or adjustments and recertification and/or verification procedures. This implies the use of various geometric reference elements: 2D plates or boards with rectangles (checkerboards) and plates with 2D circles and 3D spheres, etc.

Apart from the differences in the fields of application of each technology, one of the factors that hinders the comparison of the accuracy (and performance) of the equipment is the absence of homogeneous standards between different technologies and the lack of unified standard procedures and artefacts. In fact, 3D scanner manufacturers often use their own artefacts to test the accuracy of their equipment. Some of these artefacts may even be distributed and supplied as part of the equipment. In multiple cases, a Measurement System Analysis (MSA) is required to guarantee traceability, which implies providing the quantitative values of accuracy by statistical methods, in a procedure similar to the evaluation of measurement uncertainty in coordinate metrology devices (ISO 15530-3) [13].

The most widespread standard for evaluating such non-contact equipment is VDI/VDE 2634-2:2012 [14], which is intended for 3D optical measuring systems based on triangulation. Likewise, VDI/VDE 2634-3:2008 [15] is applied to multiple view systems based on photogrammetry. Both standards are derived from VDI/VDE 2617 for Coordinate Measuring Machines (CMM), because ISO 10360-8:2013 [16] and ISO 10360-13:2021 [17] are derived from ISO 10360 2 [18] for CMMs with contact probing.

The main idea set out in these standards [14,16] is based on 3D metrological verification with simple standard artefacts to evaluate each device independently [3,19]. In fact, these standards propose scanning a test sphere, a test ball-bar, and a test flat, among other possible artefacts. In all cases, it is mentioned that the artefact material should have a diffusely reflective surface that scatters the light in a non-volumetric manner, to enhance capturing by the 3D optical measuring system. The standard also limits the dimensions of these three artefacts (the sphere, the ball-bar, and the plane) within a range related to the measuring area/volume of the equipment. The procedure evaluates different parameters (flatness error, form probing error, size probing error, distance between sphere centres, and lengths, etc.), by comparing the calibrated values and the measured values obtained by the 3D measuring system. Calibrated values are usually obtained by contact measurement using high-precision CMMs or by other methods such as interferometry. In any case, the measurement uncertainty of the calibrated values is 4–5 times lower than the uncertainty of the acceptance or reverification procedure.

Following a similar methodology, several researchers propose the use of artefacts specifically oriented to the type of equipment to be verified, either terrestrial laser scanners or triangulation-based range sensors [20]. In the case of Gonzalez-Jorge, for example, the performance of a laser scanner is evaluated using an artefact equipped with ceramic cubes and spheres. Guidi et al. [21] compare various triangulation-based devices using different test objects (planes, cylinders, cones, and set of steps, etc.), and obtain nominal values that they compare with the CMM measurements as well as the uncertainties of the different devices within the different geometric characteristics measured. Their experimentation shows the dependence of different parameters such as resolution, noise, or dynamic range on different 3D systems. The accuracy of the CMM contact measurements (< 4 micrometres) gives sufficient traceability in this experimentation to evaluate instruments with accuracies between 20 and 300 micrometres.

In any case, it is clear that the performance, the measuring volume, and the accuracy of the 3D scanning sensor must be related to the size, morphology, and number of entities included in the artefact [22], as these define the type of verification pursued. Thus, there have even been comparisons of 3D scanners directly performed on final parts, as in the case of sculptural replicas produced by additive manufacturing [4]. Nevertheless, it is preferable to evaluate the performance of 3D scanners with free forms of a relatively high complexity, but whose dimensional accuracy is also high and can be contrasted by external calibration (with precision CMMs) [23,24]. In other cases, the comparison is oriented to the inspection of machined complex surfaces, such as those appearing in tooling (mould sector) [25,26], or it pursues a verification based on features (GD&T type) [27,28,29,30], in a similar way as in the present work.

The evolution of artefacts for metrological characterisation of non-contact 3D equipment [31] clearly suggests that colour, reflectivity, and surface finish should also be considered in the design of the standard artefact [32,33,34]. Even the possibility of using anti-reflective coatings, either projected on the surface [35,36,37] or by modifying the surface through different post-processes (sandblasting and chemical etching, etc.), has to be considered [38]. The artefact developed in the present work is an evolution of previous research work based on GD&T evaluation [27,28,29]; however, in this study it integrates not only an extended number of GD&T-based features, but also a completely new material in this field (MACOR) [39].

This material is a machinable ceramic with good mechanical and thermal properties as well as a good resistance to chemical attack (degradation). These properties, together with the possibility of achieving a good surface finish and dimensional accuracy by machining, make it suitable as a dimensional standard. In addition, MACOR parts that are machined with a high precision also have good optical properties. This last characteristic was proven in previous research [40] for different types of optical scanning equipment, from laser triangulation equipment mounted on coordinate measuring arms (AACMM) [9] to structured white light sensors [41]. In both cases, feature-based artefacts were produced by placing ceramic entities made of MACOR on CFRP (carbon fibre reinforced polymer) plates, which are light, but very stiff due to the high elastic modulus of CFRP.

Other researchers [11] also use precision spheres and ceramic gauge blocks to compare the performance of different 3D measuring systems (including Computed Tomography), but focus on the influence of the digitising technique, the point cloud density, the filtering and meshing operations, implemented by the software that manages the equipment.

More recent studies show the influence of parameters used in reverse engineering software when creating the mesh and/or CAD model from the point cloud, such as tessellation [42]. Other research explores the influence of parameters related to the scanning angles and the distance between the laser beam and the surface (in the case of laser sensors), the direction of the projection fringes (in the case of structured light equipment) [43], or the number of minimum capture shots/positions by avoiding redundant positions with particle swarm type algorithms [44,45].

These studies proved that the parameters must be optimal so that the accuracy of the adjustment provides objective and comparable values within a given technology of 3D scanning. Moreover, when the accuracies of various scanning technologies are compared, it must be considered that the results will vary depending on whether the accuracy is evaluated directly on the raw point cloud, on the filtered point cloud, or on the best-fit CAD. In the latter case, the accuracy of this adjustment must be much lower than the accuracy of the non-contact scanner, and even lower than the uncertainty of the equipment providing the reference measurements (CMM).

The present research includes the design, physical fabrication, and high-precision measurement of a new artefact with multiple geometrical features made of a machinable ceramic material. These ceramic parts provide suitable optical properties for scanning, as they are white, matte finished, and machined with high dimensional accuracy. A low surface roughness has also been achieved, but without reaching a mirror-like finish that would cause undesirable reflections. In this sense, this ceramic artefact allows for determining the capture capability of various available non-contact technologies, which are compared between them. A CMM with contact probe was used for measuring the artefact with the maximum attainable accuracy thus obtaining the reference values of the GD&T entities. With these highly accurate measurements, it was possible to generate a reference CAD model, called a “quasi-real CAD” model, which accurately reflects the real manufactured model (Figure 1b).

Four different devices of non-contact digitising technologies were evaluated: a laser triangulation sensor mounted on the CMM (LS-CMM), a laser triangulation sensor mounted on a measuring arm (LS-CMA), a fully hand-held and portable laser triangulation sensor (HLS), and a structured blue light device (LS). It should be noted that all the devices are portable, but one (LS-CMM).

The following sections describe the equipment and the methodology evaluated for their accuracy (Section 2), the steps followed in the experimentation (Section 3), the results of the dimensional and geometric analysis (Section 4), and the comparisons of the results with the different sensors (Section 5).

## 2. Materials and Methods

This section outlines the development of the ceramic artefact designed to evaluate the performance of the different reverse engineering devices considered in this study, justifying the decisions taken about the choice of material, the geometric features, and their arrangement in the artefact. Subsequently, the operations for manufacturing and assembling the different parts of the artefact are described in detail, as well as the GD&T entities that have been finally considered for the comparison. The characteristics of the 3D scanners are then introduced and finally, the methodology for comparing the 3D scanners is explained.

### 2.1. The Concept of the Ceramic GD&T Master Gauge for Metrological Evaluation

The ceramic artefact designed to carry out this research consists of different geometric features, which are included in several independent blocks, that are finally mounted on a carbon fibre base plate, 12 mm thick. Figure 1a shows the CAD model and Figure 1b shows its actual implementation. Two different ceramic materials have been employed for manufacturing the artefact blocks. On the one hand, the machinable ceramic MACOR^®^ (Corning Inc., Corning, NY, USA) [39] (SiO_2_ 46%, Al_2_O_3_ 16%, MgO 17%, K_2_O 10%, B_2_O_3_ 7%, and F 4%) was used to manufacture the prisms, cylinders, and cones, while Alumina (Al_2_O_3_) was used for the spheres. These materials have been selected because of their high density (2.52 g/cm^3^ for MACOR, 3.9 g/cm^3^ for Alumina), elastic modulus (60 GPa MACOR, 66.9 GPa Alumina) hardness (knoop100gr 250 kg/mm^2^), resistance to chemical attack, and low coefficient of thermal expansion (81 × 10^−7^ °C^−1^ for MACOR, and 72 × 10^−7^ °C^−1^ for Alumina). In the case of MACOR^®^, an additional advantage is that it can be machined by chip removal processes (turning, milling, and drilling) with high dimensional accuracy (accuracies can reach 0.013 mm in dimensions, in surface roughness an of *Ra* < 0.5 µm for the finished surface and 0.013 µm for the polished surface is achievable). In addition, its matte white colour, with no gloss, enhances point capturing by reverse engineering equipment.

Regarding the ceramic blocks (Figure 1 and Figure 2), one has been machined (CNC milling) with inclined planes at 0°, 10°, 20°, 30°, and 40° (planes denoted as PLN INC-i), and a second has had a staircase machined by defining two types of planes, some vertical (denoted as PLN V-i) and others horizontal (PLN H-i). A third block has been used to define interior features: 2 interior cylinders (CYL INT B-i) and 2 interior spheres (SPH INT B-i). In addition, 4 elements of revolution have been machined (turning and facing on a CNC lathe): 2 cylinders and 2 cones. Both the parallel cylinders (CYL EXT-1 and CYL EXT-2) were placed on a reference plane, denoted as PLN BASE, and the cones (CON-1 and CON-2) incorporated inner cylinders (CYL INT-i), and materialised distances between their upper plane (PLN SUP CYL/CON-i) and the base plane, therefore, defining the height of the elements.

All the spheres included in the artefact are precision spheres made of alumina and of commercial grade G10 according to ISO 3290-2:2014 [46] (0.25 µm form error). Alumina was chosen as the material for manufacturing the spheres for two reasons, firstly because high precision (G10) spheres were needed, and secondly, to improve the surface capture by all the compared sensors, which is severely reduced in the case of precision metal spheres. On the one hand, 3 spheres of Ø20 mm were used to align the whole artefact, and these were denoted as SPH ORIGIN O, SPH ORIGIN X, and SPH ORIGIN Y (Figure 1a). On the other hand, 5 spheres of Ø10 mm (SPH X-i) were arranged along the *X* axis and another 5 spheres of Ø10 mm (SPH Y-i) were arranged along the *Y* axis (Figure 2). Each group of 5 spheres was assembled in such a way that the theoretical line passing through their centres was inclined 10° with respect to the corresponding axis.

With all these blocks placed on the carbon fibre plate, a multitude of GD&T tolerances can be defined to perform the intended dimensional evaluation. For example, a series of dimensional and geometrical tolerances has been specified, as indicated below:Distance between spheres, between cylinders, and between parallel planes; the distance between the (parallel) cylinders was evaluated between their axes and the distance between the parallel planes was evaluated as the distance from the centroid perpendicular to the reference plane;Diameters of spheres and cylinders;Form error of spheres, planes, cylinders, and cones;Angle of inclination of planes and angle of cones, with respect to the plane XY;Parallelism between cylinders, between cones, and between planes;Perpendicularity between planes;Coaxiality between cylinders.

### 2.2. Gauge Manufacturing and Datum/Reference Measurements

Once the final design had been developed, the different elements were machined (prismatic, cylindrical, and conical blocks), the commercial spheres were acquired, and finally, all of them were assembled onto a carbon fibre base plate. Although MACOR^®^ is a machinable ceramic for industrial applications, it was necessary to develop specific strategies for clamping and cooling the workpieces, and for selecting the suitable cutting conditions in order to solve the difficulties during machining, mainly in interrupted cutting (milling) [40]. The surface finish achieved in the machined blocks was considered acceptable for the intended function of the artefact. As a reference, MACOR’s turned cylindrical surfaces achieved an average roughness of *Ra* ≈ 0.5 µm and even lower values (*Ra* ≈ 0.2 µm) were obtained in milled planar surfaces.

After machining, the ceramic blocks were adhesive bonded onto a 12 mm thick carbon fibre plate with a high elastic limit (fibres interlaced at 45°), sufficiently rigid to support all the elements with minimum deformation. This prismatic plate served as a base for all the ceramic blocks and for the carbon fibre posts that supported the ceramic spheres. On the bottom of the base plate, three conical supports were carefully distributed to minimise the structural deformation due to the weight of the artefact, which was 2.50 kg. The adhesive used to bond the different elements to the carbon fibre plate was a bicomponent epoxy (ARALDITE^®^ AW 2101), of a very high rigidity (*E* = 5500 MPa) and a minimum elongation at break (<1%), once cured. This guarantees a low structural deformation of the whole assembly.

The different entities were accurately measured (using the CMM) independently, in order to guarantee their high dimensional accuracy, even before assembly (bonding to the base plate). Their surface roughness was also assessed using a contact profilometer (roughness meter). Once all the entities were mounted on the carbon fibre base plate, all the GD&T entities established for the comparison (as described in Section 2.1) were measured by means of contact probing in a CMM. These measurements resembled a “standard” master part calibration procedure, so that the CMM measurements were traceable. Traceability is guaranteed, in first place, because the CMM was calibrated according to ISO 10360-2:2009 [18] by an external calibration laboratory, and secondly, because the uncertainty of measurements was evaluated following the guidelines of ISO 15530-3:2011 [13]. Thus, the assignment of nominals—reference values—of all the considered GD&T was carried out with a contact probe at the CMM (Figure 3a,b). In this case, a DEA Global Image 091508 CMM was used, equipped with a Renishaw SP25 scanning probe (Figure 3a) and controlled by the PC-DMIS 2018 R2 software (Figure 3b). The standard measurement procedure was performed in a room at a controlled temperature of 20 ± 0.5 °C, using contact probing strategies with a high number of points, maintaining an approximately constant density (0.5 points/mm^2^) in all features. In total, about 3800 points were probed in the different features.

The measurement program that collected the probing of features and the evaluation of the GD&T entities was executed 10 times in automatic mode to avoid the errors derived from manual probing. The standard deviation of the 10 repetitions was less than 1 µm, thus obtaining the reference values of all considered GD&T with sufficient repeatability.

### 2.3. Non-Contact Scanning Sensors

The next stage in the experimentation involved measuring the artefact with non-contact scanning sensors. During this stage, 4 laser triangulation sensors and fringe projection photogrammetry equipment were analysed (Figure 4): a laser triangulation sensor mounted on the CMM (LS-CMM), a laser triangulation sensor mounted on a coordinate measuring arm (LS-CMA), a hand-held portable laser sensor (HLS), and a structured blue light sensor (SL). Each sensor was controlled by its own software. Additionally, reverse engineering equipment is usually complemented with auxiliary materials required for its qualification (spheres and checkerboards). The CMM is a DEA Global Image 09158 (X axis: 900 mm, Y axis: 1.500 mm, and Z axis: 800 mm). Its maximum permissible error in length indication is MPE_E_ (µm) = 2.2 + 0.003·*L*, *L* in mm. For the contact measurements, a Renishaw SP25 scanning probe was used. Figure 4 shows the main characteristics of the rest of the equipment used in this research.

The captured data was managed using several metrology software packages to obtain the GD&T measurements of the different types of equipment. On the one hand, it was commented that the CMM contact measurements were programmed using PC-DMIS metrology software (version 2018 R2). On the other hand, the postprocessing of the point clouds captured by the 3D scanning systems and the subsequent GD&T evaluations were carried out with the aid of the following software packages: Geomagic Control X (2018), Polyworks, and VXelements.

### 2.4. The Comparison Methodology

The methodology used for the comparative study is outlined in Figure 5.

The ceramic artefact was measured first by contact probing in a CMM using the PC-DMIS software to generate the program and to evaluate the GD&T entities, thus providing the values that will serve as a reference for the subsequent comparative study. Subsequently, the non-contact scanning of the artefact was performed with the four available reverse engineering technologies: the Laser Triangulation Sensor on the CMM, the Laser Triangulation Sensor on a coordinate measuring arm, the Handheld Laser Triangulation Sensor, and the Structured Blue Light Scanner. Following this, the captured point clouds were processed with Geomagic Control X metrology software to obtain the evaluations of the defined GD&T entities. Finally, these values were compared with the reference values and the result of that comparison was analysed to assess the accuracy of the 3D scanning systems.

## 3. Experimental Procedure

### 3.1. Measurement of the GD&T Artefact by Contact Probing in the CMM: The Reference Measurement Values

In order to obtain the reference values of the evaluated parameters, the first scanner used for the measurement of the artefact was the CMM. The measurements of the different characteristics of this standard artefact were carried out using a calibration procedure according to ISO 15530-3:2011 [13], similar to those used on calibrated workpieces or measurement standards. This procedure is based on using calibrated workpieces or measurement standards to obtain the measurement uncertainties with the CMM. For this purpose, 10 repetitions were carried out with repositioning of the part to be calibrated (ceramic artefact). All the repetitions were performed with the same CMM contact measurement technological parameters. Specifically, the stylus used was a Ø4 mm ruby sphere on a 50 mm shank, which avoided collisions and allowed access to deep areas of the artefact. All entities could be probed with a single stylus orientation (standard orientation with rotation axes A and B in position A0B0). The origin of coordinates of the set were located at the origin sphere (SPH O) and the axes were oriented as shown in Figure 1a. The Y-axis, the long axis, was oriented along the line joining the centres of the SPH O—SPH Y spheres and oriented towards the SPH Y sphere. The X-axis, the short axis, was oriented along the line joining the centres of the SPH O—SPH X spheres and direction towards the SPH X sphere.

After the contact measurement on the CMM, a measurement of the surface roughness of the ceramic elements of the standard was also carried out. It was observed that in all cases the arithmetically measured roughness (Ra) did not exceed 2 micrometres, which is the repeatability threshold value of the CMM, ensuring a high level of measurement accuracy.

To be able to analyse the different types of equipment at a later stage, not only by evaluating the individual elements, a new CAD (quasi-real CAD) was generated. This CAD was obtained by modifying the previous one, so that the dimensions of the elements and their distances were adjusted to the real ones (contact measurements). This allowed the programming of trajectories to be adjusted to the real part in the contact measurement, as well as the evaluation of the different types of RE equipment (hardware + software) by means of the CAD comparison.

Although PC-DMIS allows the measurement results to be obtained directly from the probed points, this work modified this procedure by using the discrete point clouds to create the entities. This method of constructing entities matches the procedure for obtaining results in the Geomagic Control X software. In this way, the procedure for obtaining GD&T measurements was unified between contact (PD-DMIS) and non-contact (Geomagic Control X) probing, thus avoiding distorting the comparisons between the types of equipment.

### 3.2. Measurement of the GD&T Artefact with Non-Contact Scanning Sensors

In a second stage, all the artefact’ features were scanned with the four non-contact optical sensors: three sensors based on laser triangulation (LS-CMM, LS-CMA, HLS) and a structured blue light sensor (SL). The measurements were carried out in the same laboratory as the contact measurements and at a controlled temperature of 20 ± 0.5 °C.

#### 3.2.1. Laser Triangulation Sensor Mounted on the CMM (LS-CMM)

The scanning strategy was planned to obtain a complete coverage of the features of interest, so that several scans were performed with five different orientations of the laser head; one vertical (denoted as A0B0, according to the axis A and B of Renishaw PH10MQ motorised indexed head) and another four with the sensor inclined at 45° angle from four different orientations (Figure 6), according to axes “*X*”, “-*X*”, “*Y*” and “-*Y*” (denoted as A45B90, A45B-90, A45B180 and A45B0, respectively). Each of these orientations was pre-qualified prior to measurement with the equipment.

In all the sensor orientations the same scanning parameters were used: normal gain, a point density of 16.8 pts/mm, and a line width of 123 mm. Normal gain was selected because the surfaces of the artefact were not reflective. The chosen point density corresponds to the standard or average configuration, being also used in the qualification of the sensor. Regarding the line width, the selected value allows for capturing the largest possible area in each scan.

The other important parameters to bear in mind were those derived from the filters that were applied in real time during the scanning. As the artefact is scanned these filters were automatically applied to the generated point cloud. In particular, the angle of incidence filter discarded those captured points where the angle between the normal surface vector and the laser light direction was greater than a specific value. In this work, the angle of incidence filter was set to a value of 75°. No other filter was activated.

#### 3.2.2. Laser Triangulation Sensor Mounted on a Coordinate Measuring Arm (LS-CMA)

In this case, the sensor used was the Hexagon Metrology RS6, which was mounted on the ROMER Absolute Arm 8525-7^®^ coordinate measuring arm (Figure 7). During the manual scanning, the artefact was clamped to the granite surface plate of the CMM by means of specific fixturing devices commonly used in CMM inspection.

The parameters used for scanning the artefact with the RS6 sensor were suitable for non-reflective surfaces: point sampling 50%, exposure level 10%, exposure mode “SHINE”, angle of incidence filter of 70°, and scanning speed 300 Hz. Fine settings (line-to-line distance 1.5 mm, point-to-point distance 0.2 mm, and noise 0.025) were selected for the rest of the parameters controlled by the sensor software.

#### 3.2.3. Handheld Laser Triangulation Sensor (HLS)

The model used was the Creaforms HandyScan 700 laser triangulation sensor. In this case, as there was no external reference, it was necessary to place markers to position the artefact before scanning. For this purpose, the ceramic artefact was supported over two black plates, and a set of markers (stickers) was adhered to both the plates and the base plate of the artefact in several locations (Figure 8). The scanning parameters were as follows: the resolution or distance between points was set to 0.2 mm while the obturation was automatically adjusted to the type of surface being scanned at any one instant.

The working mode of the HandyScan 700 was a bit different from the other sensors, as it did not directly create a point cloud, but a meshed surface was generated first, and the point cloud was extracted from the vertices of the mesh afterwards. This way, better results were obtained when applying the filters implemented by the software to avoid overlapping regions in the scanned data.

#### 3.2.4. Structured Light 3D Optical Scanner (SL)

The sensor was a structured blue light portable scanner, model “3D Breuckmann smartSCAN3D-HE” (currently AICON smartSCAN^®^ from Hexagon AB, Stockholm, Sweden) mounted on a tripod. This equipment has a projection unit and an acquisition system with two cameras on each side. In the case of the structured blue light scanning, it was necessary to initially calibrate the scanner within the field of view (FOV) required for capturing the artefact. The FOV used was 125 mm, which corresponded to the smallest working field that allowed for capturing the entire volume of the piece.

The artefact was digitised in an automatic scanning mode aided by a rotary table. For the complete scanning of the artefact, 10 scans were performed, each corresponding to a 36° table rotation angle. Figure 9 shows the arrangement of the equipment during the process of capturing the point cloud in one of the 10 orientations.

### 3.3. Postprocessing of the Raw Pointclouds (Filters)

As already mentioned, in non-contact measurement equipment there are software-controlled filters that improve the scanning result by eliminating those points (known as spurious, outliers, or redundant, depending on their type) that significantly increase the error of the measurements. The main filter is the anti-overlap filter that eliminates redundant points from areas where two scans overlap. In this type of experimentation, this type of filter was applied by the software that controlled the RS6 and HandyScan scanning. Another filter applied during the point cloud acquisition was the so-called angle of incidence filter.

However, it was necessary to apply filters after capturing the point clouds and before reconstructing the geometric features and evaluating the GD&T entities defined on the artefact. Previous research has shown that the best filter for the reconstruction of features from point clouds is the so-called Sigma or Standard Deviation filter. This filter discards points that are located at a distance from the reconstructed feature further than a value that is a multiple of the standard deviation of the fit (σ, 2σ, 3σ). A study about the optimal filter was carried out, aided by the Geomagic Control X software. The methodology used for the application of the Sigma-type filter was as follows:Reconstruction of the feature from the unfiltered point cloud;Calculation of the standard deviation of the fit (point cloud to feature);Setting of the filter distance: multiplication of the standard deviation by a factor (1, 2, 3);Filtering the point cloud, removing points located at a distance superior to the previous value;Reconstruction of the feature from the filtered point cloud.

The results of this study showed that the best multiplication factor for the Sigma filter or standard deviation, was 2, which meant that the features were reconstructed from approximately 96% of the points of the corresponding point cloud. Thus, possible spurious points were removed, and the values of the evaluated parameters were within acceptable limits. The points which could cause potential errors in the GD&T measurements were largely eliminated, but without reaching an excessive point removal that could result in false measurements.

Another problem common in scanning is the impossibility of isolating the region of interest during the capturing process. The point capture was massive so that the resulting point clouds included parts of the base, the fixturing, and surfaces of the artefact not subjected to inspection, etc. Therefore, each “global” raw point cloud (Figure 9b) needed to be cleaned and clustered, by trimming operations, into the different point clouds corresponding to each feature of the artefact affected by a GD&T inspection (Figure 9c).

After separating the original point cloud, the 2 Sigma filter was applied to each individual cloud and each feature of the ceramic artefact was reconstructed using a least squares algorithm (Figure 10a). Figure 10b shows the graphical representation of the features after being generated from their corresponding point clouds, in Geomagic Control X. The GD&T measurements were evaluated from these entities to perform the metrological benchmarking.

## 4. Results

### 4.1. Dimensional Analysis

The measurement results have been collected in the form of tables and presented here in the form of graphs. For each GD&T entity evaluation, a graph shows the deviations in the non-contact measurements for the CMM contact measurements, per each geometrical feature/s involved. Thus, the performance of the four scanning sensors can be evaluated.

In the graphs included below (Figure 11, Figure 12 and Figure 13) the measurement data is represented as the deviations from the reference values obtained with the CMM. In this way, the deviations obtained in each of the GD&T measurements evaluated on the artefact are displayed together for the four devices.

In Figure 11 the largest deviations in the distances between the spheres are those obtained with the structured light (SL) sensor. The average deviation in the SL sensor for all the considered distances reaches 27 µm, while for the rest of devices the average deviations are all below 9 µm. Moreover, all the deviations corresponding to these three devices (LS-CMM, LS-CMA, and HLS) do not exceed 17 µm.

Following a similar scheme, Figure 12 shows the deviations in the diameters of these spheres, both the outer and inner ones. For the outer spheres, it can be seen that, except for the structured light (SL) sensor, which produced positive deviations, the other devices registered deviations between 0.1 and 0.2 mm below the value measured with the CMM. For the inner spheres, the hand-held laser sensor (HLS) is the one that recorded values close to those measured with the CMM. On the other hand, the LS-CMA device is the one that presents the worst data, between 0.15 and 0.20 mm.

In the case of the diameters of the outer and inner cylinders (Figure 13), the results are qualitatively similar. In other words, the measurements obtained with the structured blue light (SL) sensor also show a different behaviour from the rest of measurements, with the results closer to the reference values. This sensor provides values slightly higher than the reference diameter (CMM) in the outer cylinders and slightly lower in the inner cylinders, but below 50 µm (in absolute value) in both cases. On the other hand, it is again the laser sensor mounted on the measuring arm (LS-CMA) that records the worst data in both types of cylinders, showing deviations between 0.15 and 0.20 mm (in absolute value).

### 4.2. Form Deviations

In addition to the dimensional analysis, it is important to highlight the results obtained in terms of the form errors of the different features scanned with each of the non-contact measuring devices. The tables below show the deviations, with respect to the values obtained by contact measurements with the CMM, in the form errors for the spheres (Figure 14), for the outer and inner cylinders and cones (Figure 15), and for the different planes defined in the artefact (Figure 16). As previously calculated for the diameter measurements, the form error deviation was obtained by comparing the GD&T measurements obtained from the captured point clouds (by using Geomagic Control X) and those obtained with the CMM.

The structured light (SL) equipment is the one that records the smallest deviations in form errors for all the features. As an example, in the case of the 22 planes evaluated, 19 of them have deviations of less than 10 µm and none of the other three exceeds 23 µm. On the other hand, the most unfavourable results are recorded with the laser sensor mounted on the CMM (LS-CMM) in all the analysed features, where the deviations with respect to the reference values are in the order of tenths of a millimetre.

### 4.3. Standard Deviations in Point Clouds

Finally, a comparison was carried out focusing on the standard deviation in the point clouds with respect to the reconstructed feature. For each point cloud, this parameter is evaluated as the corrected sample standard deviation in the distances of the points to the reconstructed feature, which was obtained through fitting by applying a least squares algorithm. Therefore, it is a measure of the dispersion of the points pertaining to a point cloud regarding the feature that has been reconstructed from the point cloud, or, in other words, it is a measure that allows for characterising the quality of the point cloud.

The three graphs presented below (Figure 17, Figure 18 and Figure 19) clearly show how the standard deviation values for the four devices are below 30 µm. More specifically, for the laser sensor mounted on the arm (LS-CMA), the hand-held laser (HLS), and the structured light (SL) devices, the values below 18 µm are recorded. Once again, the structured blue light (SL) sensor provides higher quality point clouds with respect to the best fitting features.

The standard deviation values for both the inner and outer spheres (Figure 17) are quite homogeneous, lying in a narrow range for each device. In fact, the largest oscillation occurs for the laser sensor mounted on the arm (LS-CMA) where a difference of only 10 µm is found (18 µm for SPH X-4 and 8 µm for SPH ORIGIN Y).

In the case of both inner and outer cylinders and cones (Figure 18), a remarkable uniformity is also observed. The laser sensor mounted on the CMM (LS-CMM) is the device that produces the maximum difference, 11 µm, in the measurements of the outer cylinders.

Finally, Figure 19 shows the data referring to the standard deviation in the point clouds corresponding to the planes. In this case, the most significant data are the values achieved with the LS-CMM sensor in the inclined planes. While the values are quite homogeneous in the horizontal and vertical planes, in the inclined planes the standard deviation increases as the inclination increases. Nevertheless, the worst value, reached in the plane with the highest inclination (40°), is below 30 µm, which endorses the goodness and quality of the point cloud captured with the LS-CMM sensor.

### 4.4. Determination of the Uncertainty of Measurements

This section is devoted to providing an estimation of the measurement uncertainty that can be reasonably assigned to the different measurements made with the four devices considered. Although there are specific standards for the acceptance and reverification (maximum permissible errors) of some of the equipment considered (such as the LS-CMM and the LS-CMA laser triangulation equipment), these standards are not applicable to others (such as the HLS sensor or the SL sensor). Therefore, the generic recommendations of the documents EA-4/02:2002 [47] and the ISO 15530-3:2011 [13] have been followed. The latter allows the homogenisation of the uncertainty calculation for the four sensors, based on comparing their measurements to a calibrated artefact, which will be used as a standard.

The first step is to assign uncertainties to the different entities of the ceramic artefact, during the artefact calibration with the CMM. The artefact was measured 10 times in the optimal working space of the CMM, in a temperature-controlled lab (20 ± 0.5 °C) in a short interval of time, obtaining 10 measurements of all the entities of interest. Moreover, the CMM thermal compensation was used to replicate the same procedure applied during the last CMM calibration. Therefore, it can be considered that the influence of the temperature is already included in the uncertainty associated with the CMM ($u_{CMM_{j}}$), which was estimated by considering the limits of the maximum permissible error. 

Table 1 shows the uncertainty budget corresponding to the artefact measurements, that includes 3 contributions: (1) uncertainty due to the CMM measurements ($u_{cal_{j}}$) in the working volume, (2) uncertainty due to the repeatability of the measurements ($u_{rep_{j}}$), which is calculated from the standard deviation in the CMM measurements ($S_{CMM_{j}}$), and (3) uncertainty to the reproducibility of the measurements ($u_{REP_{j}}$). This results in a maximum value of the expanded uncertainty for the artefact entities’ measurements ($U_{g_{j}}$). In this sense, the maximum value of all of them has been obtained for the uncertainty in the measurement of the distance between the farthest spheres (alignment spheres), reaching 0.0026 mm. 

Regarding the assignment of uncertainties of the measurements performed with each of the 3D sensors on each GD&T entity (dimensions and form error or standard deviation of the best fit cloud), Table 2 shows the uncertainty budget, based on the four main contributions. The first contribution is due to the repeatability of the measurements carried out with each sensor ($u_{rep3d_{j}}$), and has been determined from the standard deviation $S_{j}$ of the differences between the CMM measurement (reference) and the measurements of each sensor for each evaluated entity. For this contribution, the values of three measurements (n=3) performed with each sensor on the ceramic artefact have been considered. The second contribution is due to the maximum differential expansion during the test, or uncertainty due to the temperature variation ($u_{\Delta T_{j}}$). A rectangular distribution has been considered for this uncertainty, with a maximum thermal difference of ΔT=1 ℃, being the Coefficients of Thermal Expansion (CTE) of $9\cdot 10^{-6}\;K^{-1}$ for MACOR^®^ and $6\cdot 10^{-6}\;K^{-1}$ for Alumina, and being $L_{j}$ as the maximum dimension to measure on each entity considered. The third contribution is due to the resolution E of each optical sensor ($u_{E}$). In this case, a value E=0.001 mm is assigned as the most conservative value, although for the analysis of the different entities, point cloud coordinates have been considered with a number of decimal places higher than three. Finally, the fourth contribution is the uncertainty assigned to the reference measurements ($u_{g_{j}}$) performed during the artefact calibration with the CMM (Table 1). In this sense, the largest of the uncertainties obtained in the artefact calibration has also been taken as the most conservative value.

Table 3 shows, as an example, the expanded uncertainties for some of the most representative GD&T entities of the artefact, considering a coverage factor k=2. It should be noted that many of the factors considered have been overestimated, so that the values indicated in the table represent the highest value of the various GD&T entities considered in each row.

## 5. Comparison and Discussion of Results

The measurement results aggregated by the GD&T entities, such as dimensions and form errors for different types of features (spheres, cylinders, cones, and planes, etc.), also distinguishing between exterior and interior features, allows the discussion of results. It should be noted that, due to the inherent differences in each type of equipment (manual or automatic operation, application of filters during digitising, and measurement parameters, etc.), it was necessary to define a common methodology and to determine a consistent value for the Sigma filter applied to the captured point clouds (2S).

Considering all the measurements made with the different types of equipment and the GD&T parameters analysed, it is possible to differentiate the results obtained from the two technologies used in the research (laser triangulation and structured light). Structured light offers appreciably better values (closer to the reference ones) than the rest of the equipment in all the analyses, both by entities and by parameters, except in the case of the distances between spheres (Figure 20), highlighting the dependence of the SL equipment on the distance between the entities to be measured. A smaller working range would be required in this case.

If diameter is the GD&T specification to evaluate and compare (Figure 21), the structured light (SL) technology provides the closest values to the reference ones, although in this case the average deviation is limited by values of up to 50 µm. In contrast, the equipment based on triangulation laser technology does not achieve the acceptable values, as the average deviations are in the order of magnitude of tenths of a millimetre.

The structured light (SL) equipment is also the best technological choice in terms of the geometrical accuracy of the single features (the average of the form error deviations of all features does not exceed 15 µm). Figure 22 also shows that the laser triangulation sensor mounted on the CMM (LS-CMM) produces the largest deviations in the form error value. In this case, the values obtained are much higher than those acceptable for all the features measured on a reference artefact made of ceramic materials (Alumina and MACOR^®^).

It should be noted that the largest deviations in the measurements were found, regardless of the equipment, in the form error of the outer spheres, precisely those that were made of a shiny material (Alumina Spheres) instead of the rest of matt finish features (MACOR^®^).

Regarding the uncertainty evaluation, a comprehensive analysis has been performed based on calibrating all entities and the GD&T characteristics of the artefact, propagating such uncertainty to the uncertainty of each sensor measurement. The values reflected in Table 3 are a summary of the maximum expanded uncertainties attributed to the grouped entities. 

Although the uncertainty values appear high, two effects must be considered. On the one hand, the uncertainty values (Table 3) reflect the maximum values of all differences between them (per row, considering all cylinders, all spheres, and all planes, etc.). On the other hand, the uncertainty values should be compared with the nominal values of the different GD&T parameters, and not with the differences between the devices, as these differences are quantitatively lower.

## 6. Conclusions

The analysis of the state of the art of non-contact reverse engineering equipment has revealed that there is a need for calibration artefacts together with “unified” verification procedures that allow for assessing the metrological performance of this type of equipment. This is partly because there are many scanning technologies, many reverse engineering software packages (with subtle differences in filtering methods and capture parameters, etc.), and very different reference geometries and materials. Hence, the solution adopted in this work is based on the development of an artefact consisting of machinable blocks made of a ceramic material (MACOR^®^) that provides good optical properties (non-reflective surface), a low coefficient of thermal expansion, a high hardness, the resistance to corrosion, stability, and ease of cleaning, etc. Moreover, this material is machinable thus ensuring a good surface finish and dimensional accuracy of the features included in the artefact.

In this research, the optical behaviour of the described artefact consisting of different geometrical features made of ceramic materials is analysed. The typology, number, and distribution of the features have been specifically designed to be suitable for the verification of four non-contact scanning devices. By scanning the artefact, it is possible to perform a “multi-feature” evaluation of a broad range of GD&T specifications, which constitutes an evolution of the current procedures proposed in the VDI/VDE and ISO standards. 

This study analyses the measurement performance of four non-contact 3D scanners, three of them based on the laser triangulation principle (LS-CMM, LS-CMA, and HLS), and one based on structured light technology (LS). All of the scanners are portable, but the one which is mounted on a CMM (LS-CMM). Moreover, two of them (LS-CMM and HLS) operate automatically, while the other two (LS-CMA and HLS) are manually operated during scanning.

The study generates benefits for different types of users of reverse engineering equipment. In the case of a calibration laboratory, the artefact provides them with new possibilities to propose calibration solutions for non-contact equipment. In the field of reverse engineering, an artefact like the one proposed helps to extend the capabilities of the reverse engineering equipment (commonly dedicated to the reconstruction of damaged parts, and heritage and art conservation, etc.), to sectors like mechanical design and manufacturing inspection. In these cases, the inspection or verification of dimensional and geometric tolerances requires providing traceability to the results of the performed measurements. Ultimately, focusing on a particular scanning device, the artefact developed in this research allows for determining the degree of agreement between the non-contact measurements performed with the device and the actual reference values, obtained previously by contact probing in a CMM. This way, the measurements derived from the point clouds captured by the scanning device are traceable and, moreover, subject to reliable comparisons with measurements performed by other non-contact scanners.

## Figures and Tables

**Figure 1 sensors-22-08596-f001:**
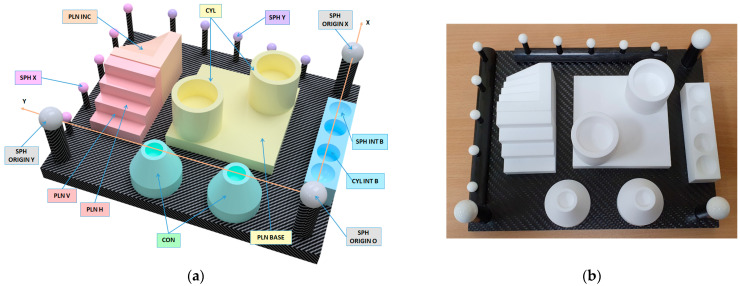
The artefact used in the benchmarking: (**a**) CAD model of the ceramic artefact. (**b**) Manufactured ceramic artefact.

**Figure 2 sensors-22-08596-f002:**
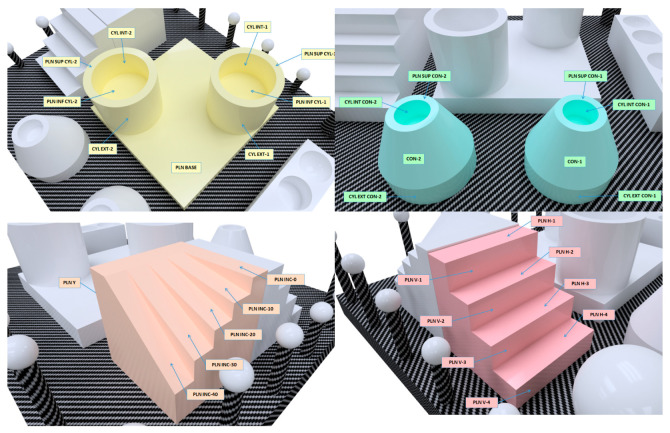
The nomenclature and designation of the geometric features of the artefact.

**Figure 3 sensors-22-08596-f003:**
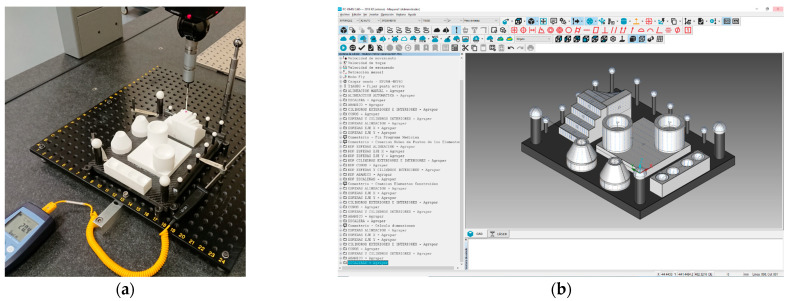
CMM measurement of the ceramic artefact: (**a**) Detail of the artefact in the CMM and contact probing with Renishaw SP25. (**b**) Measurement program in PC-DMIS software.

**Figure 4 sensors-22-08596-f004:**
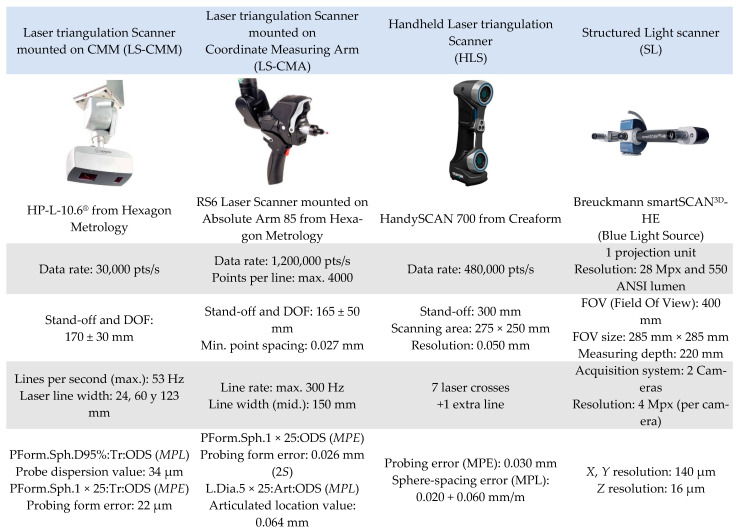
Available scanners and their main technical features.

**Figure 5 sensors-22-08596-f005:**
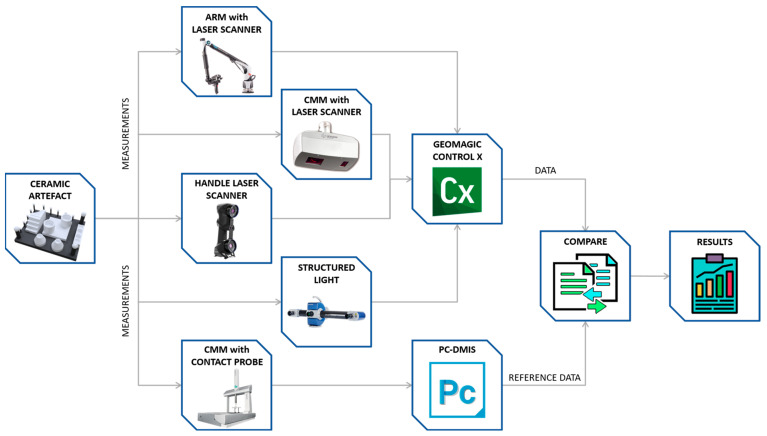
Methodology for comparing the measurement results.

**Figure 6 sensors-22-08596-f006:**
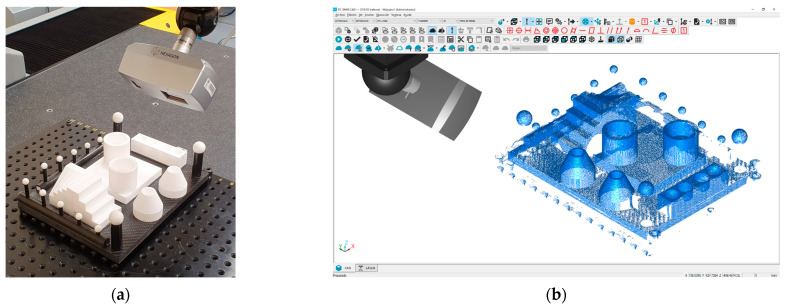
(**a**) Scanning with HP-L-10.6 sensor on the CMM, at an orientation of A45B0 (45° regard -Y axis). (**b**) An example of the raw point cloud captured with the five sensor orientations.

**Figure 7 sensors-22-08596-f007:**
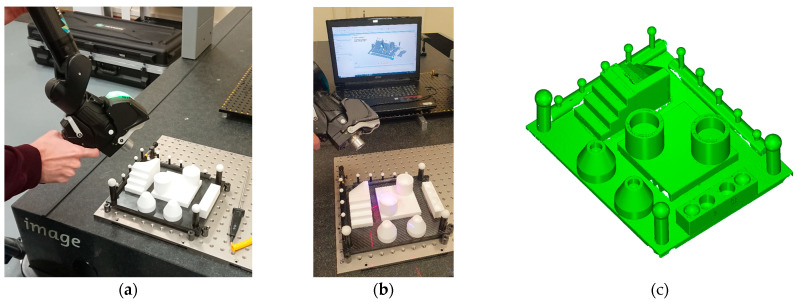
Scanning the ceramic artefact with the RS6 laser sensor mounted on the Romer Absolute Arm 8525-7^®^: (**a**,**b**) Execution of the scanning; (**c**) Result of the scanning.

**Figure 8 sensors-22-08596-f008:**
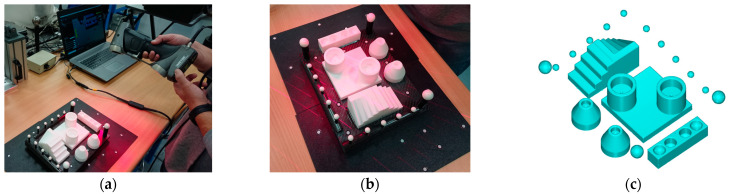
Scanning the ceramic artefact with the HandyScan 700 sensor. (**a**,**b**) Execution of the scanning; (**c**) Result of the scanning.

**Figure 9 sensors-22-08596-f009:**
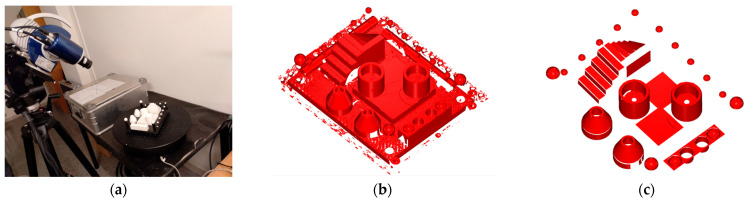
(**a**) Scanning the artefact with the Breuckmann smartSCAN3D-HE sensor (SL). (**b**) Example of a captured raw point cloud. (**c**) Point cloud after trimming, including only the features of interest.

**Figure 10 sensors-22-08596-f010:**
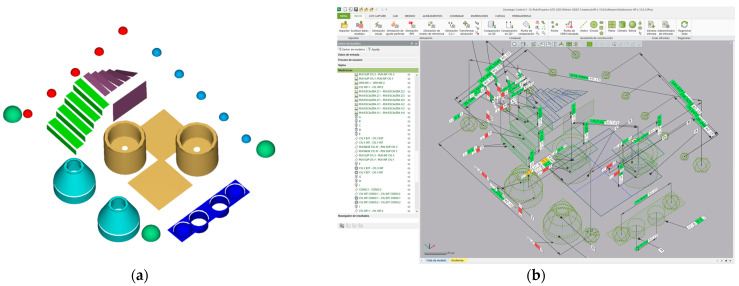
The GD&T measurements of the ceramic artefact. (**a**) Best-fit features as reconstructed from point clouds. (**b**) Measurement of GD&T entities on Geomagic Control X.

**Figure 11 sensors-22-08596-f011:**
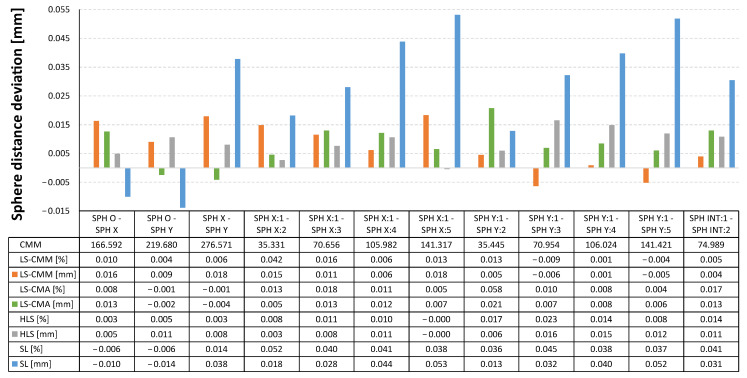
Deviations in the distance between the spheres: non-contact measurements vs. contact CMM measurements.

**Figure 12 sensors-22-08596-f012:**
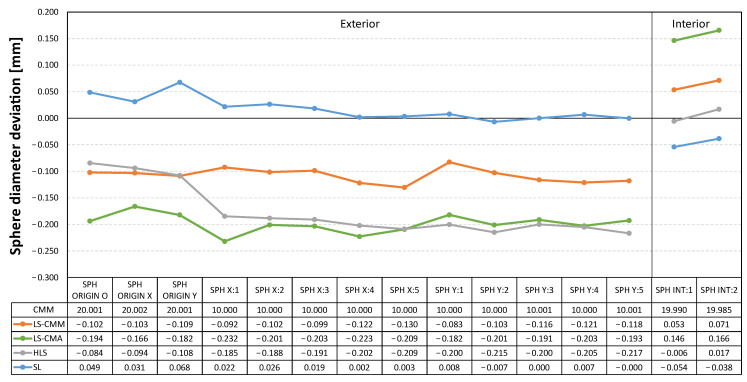
Deviations in the sphere diameters: non-contact measurements vs. contact CMM measurements.

**Figure 13 sensors-22-08596-f013:**
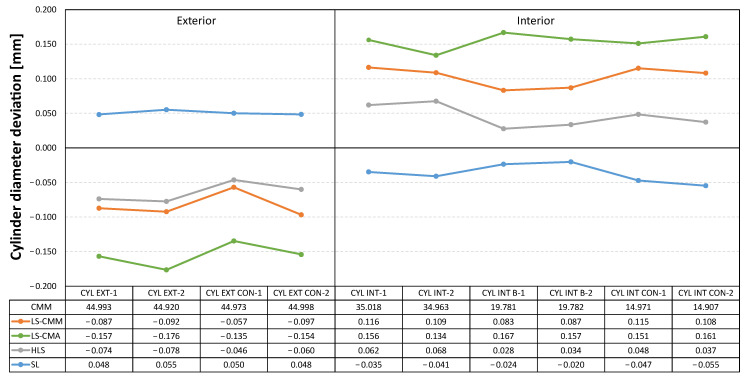
Deviations in the cylinder diameters: non-contact measurements vs. contact CMM measurements.

**Figure 14 sensors-22-08596-f014:**
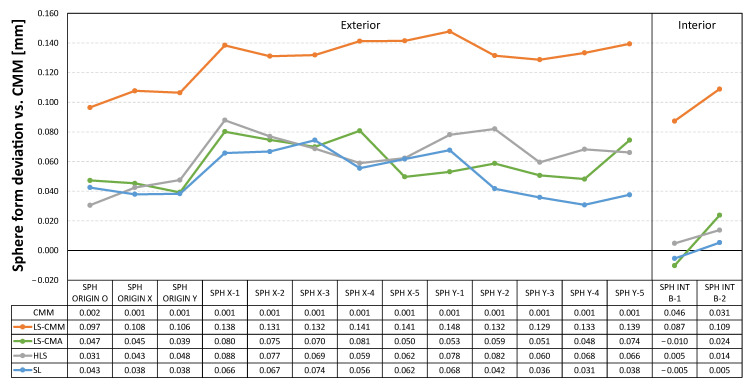
Deviations in the form error for the spheres (sphericity): non-contact measurements vs. CMM contact measurements.

**Figure 15 sensors-22-08596-f015:**
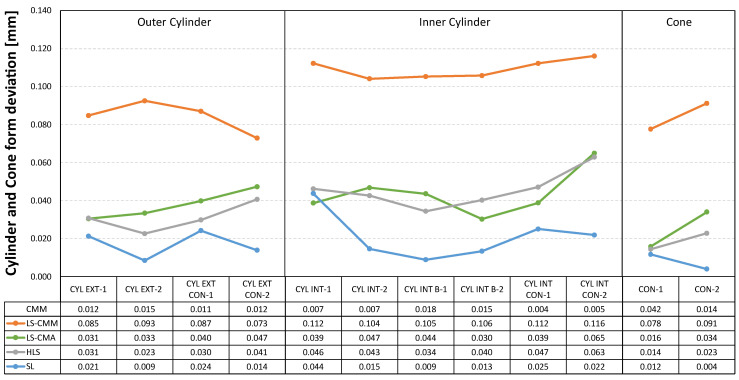
Deviations in the form error for the cylinders (cylindricity) and cones: non-contact measurements vs. CMM contact measurements.

**Figure 16 sensors-22-08596-f016:**
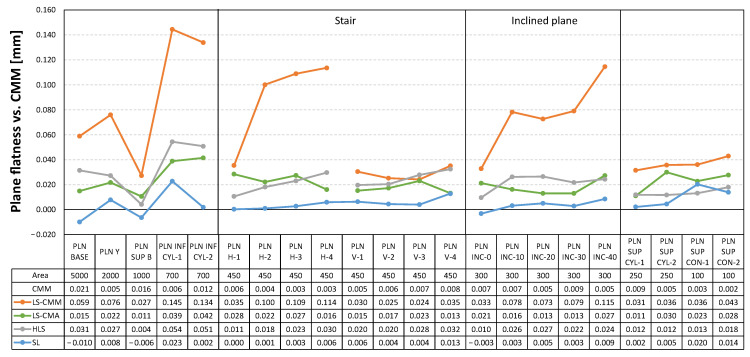
Deviations in the form error for the planes (flatness): non-contact measurements vs. CMM contact measurements.

**Figure 17 sensors-22-08596-f017:**
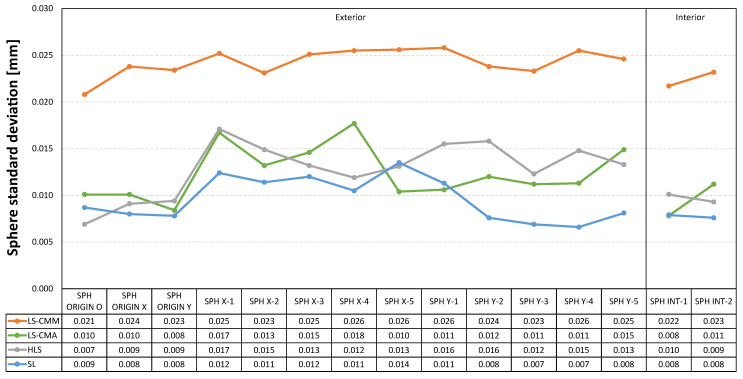
Standard deviation in the point clouds corresponding to the spherical features.

**Figure 18 sensors-22-08596-f018:**
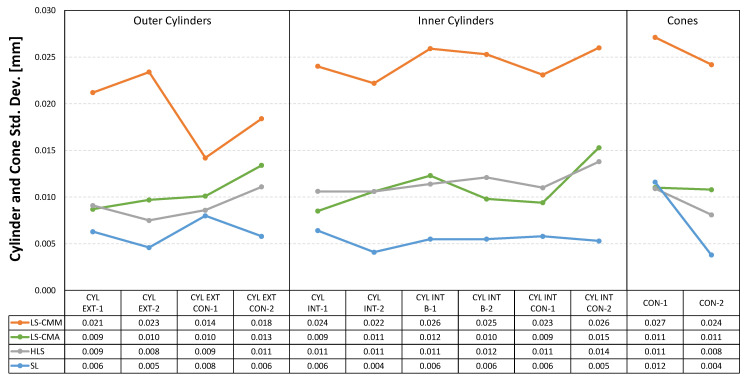
Standard deviation in the point clouds corresponding to the cylindrical and conical features.

**Figure 19 sensors-22-08596-f019:**
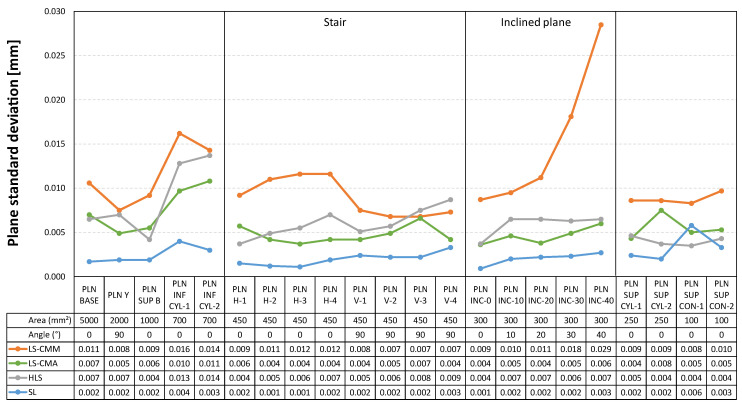
Standard deviation in the point clouds corresponding to the planar features.

**Figure 20 sensors-22-08596-f020:**
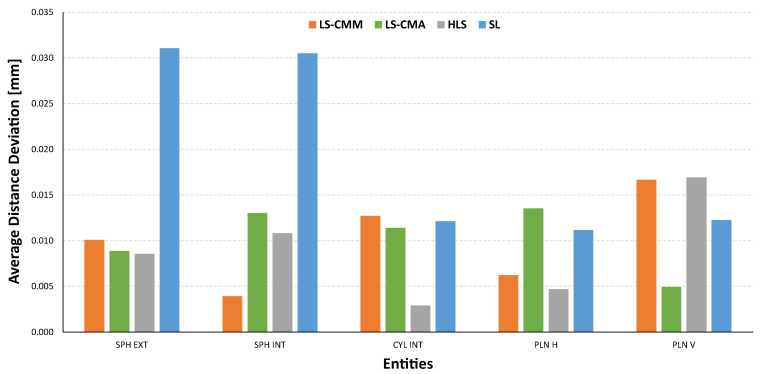
Comparison of the aggregated average deviations regarding the distance between features.

**Figure 21 sensors-22-08596-f021:**
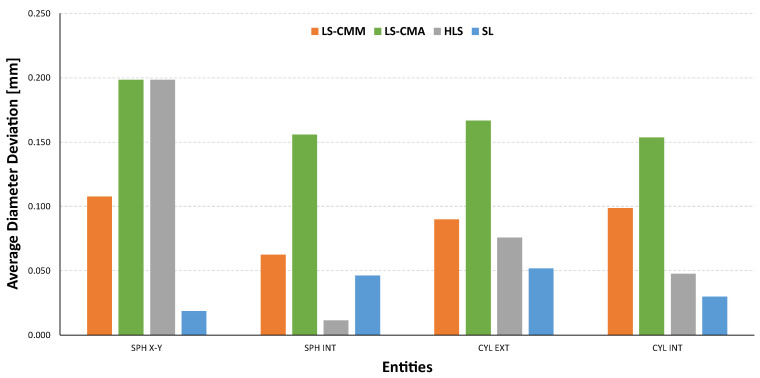
Comparison of the aggregated average deviations regarding the sphere and cylinder diameters.

**Figure 22 sensors-22-08596-f022:**
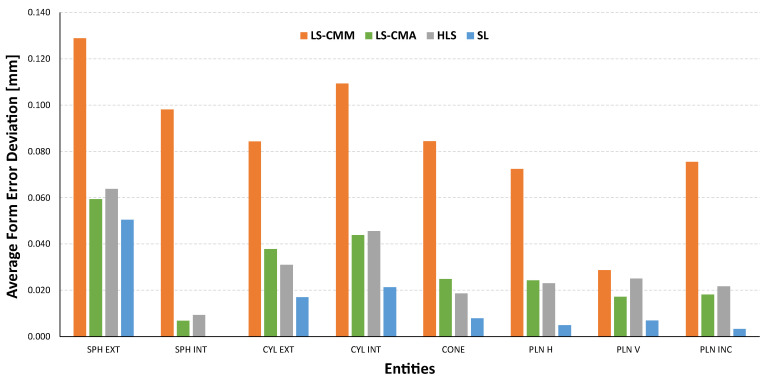
Comparison of the aggregated average deviations regarding the form error of features.

**Table 1 sensors-22-08596-t001:** Uncertainty budget for the measurement of artefact entity *j*.

Contribution		Eval. Type	Distribution	Sensitivity Coef.	Standard Uncertainty
Uncertainty of CMM used for calibration	$u_{cal_{j}}$	B	Rectangular	1	$\sqrt{\frac{1}{12}{\left(u_{CMM_{j}}\right)}^{2}}$
Repeatability of measurements during calibration with CMM	$u_{rep_{j}}$	A	Normal	1	$\sqrt{{\left(\frac{S_{CMM_{j}}}{\sqrt{m}}\right)}^{2}}$
Reproducibility during calibration with CMM	$u_{REP_{j}}$	B	Rectangular	1	$\sqrt{\frac{1}{12}{\left(u_{CMM_{j}}\right)}^{2}}$
		Combined Standard Uncertainty			$u_{g_{j}}=\sqrt{u_{cal_{j}}^{2}+u_{rep_{j}}^{2}+u_{REP_{j}}^{2}}$
		Coverage Factor k			2
		Expanded Uncertainty			$U_{g_{j}}=k\cdot u_{g_{j}}$

**Table 2 sensors-22-08596-t002:** Uncertainty Budget for the 3D sensor measurements of artefact feature *j*.

Contribution		Eval. Type	Distribution	Standard Uncertainty
Measurement Repeatability	$u_{rep3d_{j}}$	A	Normal	$\frac{S_{j}}{\sqrt{n}}$
Variation of temperature	$u_{\Delta T_{j}}$	B	Rectangular	$\frac{CTE\cdot L_{j}\cdot 2\cdot \Delta T}{\sqrt{6}}$
3D Sensor Resolution	$u_{E}$	B	Rectangular	$\frac{E}{\sqrt{12}}$
Artefact calibration	$u_{g_{j}}$	B	Stated in previous calibration	$u_{g_{j}}$
			Expanded uncertainty	$U_{3D\;sensor}=k\;.\sqrt{u_{rep_{j}}{}^{2}+u_{g_{j}}{}^{2}+u_{\Delta T_{j}}{}^{2}+u_{E}{}^{2}}$

**Table 3 sensors-22-08596-t003:** Example of several expanded uncertainties values (in mm) for the 3D sensors’ measurements.

**GD&T Type**	**GD&T Feature**	$U_{LS-CMM}$	$U_{LS-CMA}$	$U_{HLS}$	$U_{SL}$
Diameters	SPH (external)	0.0109	0.0107	0.0086	0.0086
	CYL EXT	0.0125	0.0120	0.0104	0.0056
	CYL INT	0.0105	0.0088	0.0114	0.0099
Distances	SPH X-Xi & SPH Y-Yi	0.0078	0.0061	0.0065	0.0109
	CYL EXT & CYL INT	0.0068	0.0100	0.0059	0.0089
	PLN H-Hi &PLN V-Vi	0.0103	0.0087	0.0097	0.0064
Form Errors	SPH (all)	0.0114	0.0108	0.0114	0.0108
	CYL EXT & CYL INT	0.0258	0.0092	0.0114	0.0114
	PLN H & PLN V	0.0258	0.0064	0.0068	0.0058
	CON EXT & CON INT	0.0191	0.0108	0.0125	0.0068
Point clouds Std Dev (Best fit)	SPH (all)	0.0052	0.0054	0.0053	0.0054
	CYL EXT & CYL INT	0.0057	0.0054	0.0054	0.0053
	PLN H & PLN V	0.0054	0.0052	0.0053	0.0052
	CON EXT & CON INT	0.0054	0.0052	0.0054	0.0063

## Data Availability

Not applicable.

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
