# Peer review of "Metrology Benchmarking of 3D Scanning Sensors Using a Ceramic GD&T-Based Artefact"

_sensors, 2022, doi:10.3390/s22228596_

Round 1

Reviewer 1 Report

The authors present an interesting paper, but the manuscript in its present form needs more elaboration. First of all, I consider that the manuscript does not meet the definition of ‘Article’ as it is a technical work, therefore, its type must change from ‘Article’ to ‘Technical note’.

Secondly, the manuscript lacks a discussion as expected in a scientific manuscript. Section 5 is various tables and the text does not provide additional insights into the causes. When there is a large discrepancy, the authors report it but do not discuss it. From my point of view, the manuscript is incomplete and requires a proper discussion section. Despite the interesting work and the developed artifact, I consider that the manuscript has an unacceptable deficiency in its structure.

Figure 1 needs to be reworked and state the same information per row. Authors have to report the uncertainty (and the sigma level), the measurement system, spatial resolution …etc for all the instruments. Besides, the authors report a 26 microns ‘accuracy’ at 2 sigmas for the RS6 alone, but it is only a part of the final error component, as the absolute arm contributes with a higher component. This has to be revised in depth. Moreover, the authors have to revise the terminology, since they are stating ‘accuracy’, but it seems to be more appropriate ‘precision’ or even ‘uncertainty’. This is very important in metrology.

In line 441, the authors stated that the HandyScan 700’s point cloud is extracted from the original mesh, but they did not provide any additional info. How it was done? Using the vertexes of the triangles?

Lines 505-514 and Figure 11 do not provide any meaningful information. Authors have to report the point density (superficial and/or volumetric) or remove them.

In section 4.2, the authors have to report clearly what they considered a ‘deviation’ (see line 564) to understand better the results. It is a mere discrepancy value? Or is it the result of a specific formula?

In line 628 the authors declare that the ‘Breukmann’ is the worst of the four. Since it employs a structure light measurement system, and the authors only considered one specific protocol for data acquisition, it is very bold to claim it. It has to be rewritten to that into account the dependency of the data acquisition protocol followed.

In lines 652-654 appear large deviations for the spheres made of a different material. This calls into question the appropriateness of these components of the developed artifact and needs to be discussed in a clear and unbiased manner.

Reviewer 2 Report

Dear authors,

In the beginning, I would like to thank you for updating the paper. However, there are some issues that I want to point out in order to improve the paper. The quality of the information is especially important for a technical review type of paper (technical benchmarking).

A)    General remarks

1.      In the case of the title and the abstract, those elements must be understandable to none familiar to the field reader. It is advised to use no abbreviations if possible. The reviewer suggests at least not using the GD&T abbreviation in the title so the title is clear. In the abstract, the authors are using abbreviations, which should be avoided in the abstract and those abbreviations should be introduced for the first time in the main text.

3. Authors are using conditional forms extensively which should be avoided in scientific text. Phrases like “can”, “could”, and “should” should be avoided if possible. This unjustified use is especially visible in some parts of the paper. A scientific paper should be written so that only definite statements are presented to the reader.

5. The introduction part looks good and can meet the criteria for a proper introduction. Also, the literature review of this chapter is mostly correct. However, the reviewer would suggest a wider state-of-art review in the case of optical, none-contact methods used in metrology and mechanical engineering in general. Although the authors are focusing later on geometrical measurements, the reviewer would suggest at least mentioning other types of measurements performed by the optical systems like:

·         The most important one, which is not mentioned, is 3D Laser Doppler Vibrometry. The primary usage is vibration measurements of structures and systems. Here is an example of lightweight truss measurements Guinchard, M. et.al Non-invasive measurements of ultra-lightweight composite materials using laser Doppler Vibrometry system, Proceedings of the 26th International Congress on Sound and Vibration, ICSV 2019. Another application is for quality control which can find applications also for RC structures e.g DOI: 10.1109/IDAACS53288.2021.9661060

·         This method can be also used for strain-stress measurements from the data measured or if the system has special instrumentation for direct stress-strain measurements. E,g of applications can be found here:  DOI: 10.1007/s11340-011-9545-5, https://doi.org/10.3390/s21010319 and many others. The system is also widely used for crack detection or delamination problems e.g https://doi.org/10.1016/j.optlaseng.2016.10.022 .

·         Also, Digital Image Correlation, which is mostly used for strain/stress measurements can be also used for geometrical inspections like here https://doi.org/10.1117/1.OE.58.9.094105

It is suggested to include this information in this chapter. Otherwise, this chapter is interesting and maybe it could be also improved with some graphical information on these methods, equipment or applications of these techniques.

6.  The methods are described very well. No changes, in the reviewers' opinion, are necessary at this point.

7. Suggest changing the name of the chapter “Experimentation” which is not fully professional (e.g experimental validation, experimental setup, etc.). Otherwise, this chapter is also very clear and no specific errors in the methodology were detected.

8. Results description, evaluation and comparison as well as final conclusions are well prepared and no problems were detected.

B)    Item remarks

The paper's graphical input is very good. No significant changes are required.

C)    Conclusions

Due to some improvements, the reviewer is marking the paper for minor revision and will happily read the final version.

Reviewer 3 Report

The manuscript is devoted to comparing the metrological characteristics of 3D scanning sensors. To make comparisons between 3D scanning devices, a special artifact equipped with several ceramic objects has been developed. Extensive experimental researches are presented. At the same time, the analysis of the results was limited to a simple statement of the data obtained. Moreover, the purpose of the research is not entirely clear. Calibration procedures and standards for scanning measuring devices are standardized. How does the new artifact enhance the ability to calibrate measuring devices or analyze measurement data? In addition to this remark, there are other less significant remarks.

1. The developed artifact must be compared with the known ones and its advantages should be shown.

2. It is desirable to provide a calculation of the measurement uncertainty.

3. It is advisable to show the effect of scanning modes on the measurement error.

4. It is desirable to supplement the comparative analysis of four 3D scanning sensors with data on the time of scanning and processing the results.

5. The conclusion should have significant quantitative findings.

Round 2

Reviewer 1 Report

The authors have done a good job of revision and have significantly improved the paper, especially section 5 (apart from Figure 4 and other subsections).

However, two issues remain:

A major issue in the relation to the manuscript type. I maintain my initial position because despite the authors' answer stating that no technical note has been published in Sensors and the submission cannot be classified as it; in fact, if the authors check the journal they can see that this year (2022) 14 technical notes were published, some of them with more than 20 pages (https://doi.org/10.3390/s22187049) or even 30 pages ( https://doi.org/10.3390/s22176574)

A minor issue, in relation to the un-meshing of HandyScan 700. The authors have to state their answer in the manuscript so that the reader has no doubts. I checked lines  445 - 449 and I was unable to locate it.

Reviewer 3 Report

The manuscript of the article has been revised. Changes were made to the manuscript based on the comments of the reviewer. All this allows us to conclude that the manuscript has changed for the better. Therefore, I consider it possible to recommend the manuscript for publication.

Author Response

Thank you very much for your invaluable contribution, it has helped to improve our article considerably.